# Molecular Mechanisms of Antiproliferative Effects of Natural Chalcones

**DOI:** 10.3390/cancers13112730

**Published:** 2021-05-31

**Authors:** Radka Michalkova, Ladislav Mirossay, Maria Gazdova, Martin Kello, Jan Mojzis

**Affiliations:** Department of Pharmacology, Faculty of Medicine, Pavol Jozef Šafárik University, 040 01 Košice, Slovakia; radka.michalkova@student.upjs.sk (R.M.); ladislav.mirossay@upjs.sk (L.M.); maria.gazdova@student.upjs.sk (M.G.); martin.kello@upjs.sk (M.K.)

**Keywords:** chalcones, cell death, apoptosis, autophagy, cell cycle arrest, angiogenesis, signaling pathway

## Abstract

**Simple Summary:**

Despite the important progress in cancer treatment in the past decades, the mortality rates in some types of cancer have not significantly decreased. Therefore, the search for novel anticancer drugs has become a topic of great interest. Chalcones, precursors of flavonoid synthesis in plants, have been documented as natural compounds with pleiotropic biological effects including antiproliferative/anticancer activity. This article focuses on the knowledge on molecular mechanisms of antiproliferative action of chalcones and draws attention to this group of natural compounds that may be of importance in the treatment of cancer disease.

**Abstract:**

Although great progress has been made in the treatment of cancer, the search for new promising molecules with antitumor activity is still one of the greatest challenges in the fight against cancer due to the increasing number of new cases each year. Chalcones (1,3-diphenyl-2-propen-1-one), the precursors of flavonoid synthesis in higher plants, possess a wide spectrum of biological activities including antimicrobial, anti-inflammatory, antioxidant, and anticancer. A plethora of molecular mechanisms of action have been documented, including induction of apoptosis, autophagy, or other types of cell death, cell cycle changes, and modulation of several signaling pathways associated with cell survival or death. In addition, blockade of several steps of angiogenesis and proteasome inhibition has also been documented. This review summarizes the basic molecular mechanisms related to the antiproliferative effects of chalcones, focusing on research articles from the years January 2015–February 2021.

## 1. Introduction

Despite the significant progress in the field of anticancer therapy, the mortality rates in some types of cancer have not significantly decreased in the past decades [1]. Moreover, clinical use of anticancer agents is often limited by severe organ toxicity [2,3]. Therefore, the search for novel therapeutic modalities, such as compounds from natural sources, has become a topic of great interest.

Plants have been known for their effects on the human body since the beginning of human history [4]. The Ebers papyrus, written around 1500 BC, is one of the most famous ancient records of the use of plants to cure diseases [5]. For cancer research, nature is an important source of active substances as well as a scaffold for the development of their semisynthetic or synthetic derivatives with antitumor effects [6,7,8].

Epidemiological observations suggest that high consumption of vegetables and fruits reduces the incidence of several types of tumors in humans [9,10]. One of the greatest group of phytochemicals with a broad spectrum of biological activities is polyphenols. A number of articles focusing on the anticancer effect of polyphenols have been released in the past decades [11,12,13,14]. Among them, chalcones, the precursors of flavonoid biosynthesis in higher plants, attract attention for a wide spectrum of biological actions with clinical potential [15].

Chemically, chalcones are derivatives of aromatic ketones, 1,3-diphenyl-2-propen-1-ones. Based on the substituents, they are classified as hydroxychalcones, methoxychalcones, aminochalcones, arylchalcones, alkylchalcones, nitrogenous chalcones, and others [6]. The simple chemical structure (Figure 1) and a wide range of modifications in their molecules increase the ability of chalcones to affect various molecular and cell signaling targets in the organism [8]. Plenty of studies have documented numerous biological actions of chalcones including anti-inflammatory [16], antirheumatic [17], antidiabetic [18], antimicrobial [19,20,21], immunomodulatory [22], antimalarial [23], antiparasitic [24], and antiproliferative [25,26] effects. Moreover, the anticancer effect of chalcones in experimental animals has also been shown [27,28]. Additionally, in contrast to “classical” anticancer agents, chalcones are well tolerated as documented in animal as well as clinical studies [29,30,31,32,33,34].

The aim of this paper is to summarize the current knowledge about the basic molecular mechanisms of antiproliferative actions of natural chalcones, using scientific literature from the years January 2015–February 2021.

## 2. Induction of Cell Death

The goal of conventional therapy is to suppress cell proliferation and to induce cancer cell death. Except “classical” types of cell death such as apoptosis and necrosis, there are multiple mechanisms of cell death known today [35]. In the following, we describe the mechanisms of action of chalcones in relation to apoptotic and non-apoptotic cell death induction.

### 2.1. Induction of Apoptosis

The insensitivity of tumor cells to proapoptotic stimuli is one of the basic cancer hallmarks [36]. Suppression of apoptosis leads to longer cell survival and thus increases the risk of mutation accumulation, resulting in increased invasiveness, stimulation of angiogenesis, deregulation of the cell cycle, and other processes associated with tumorigenesis. Cancer cells are able to avoid apoptosis by various mechanisms that result in deviation from normal prosurvival and proapoptotic regulation [37]. A major role in the suppression of apoptosis is associated with the loss of function of key proteins involved in the regulation of apoptosis, particularly by increasing the levels of antiapoptotic molecules including Bcl-2, Mcl-1, HSP90, survivin, and others and the downregulation or mutation of proapoptotic proteins such as BAX and BAK [38]. Many chalcones have been shown to be capable of causing cell death through the induction of mitochondria, receptor, and/or endoplasmic reticulum-mediated apoptosis. The results of current in vitro studies are presented below.

#### 2.1.1. Intrinsic Pathway of Apoptosis

Intrinsic stimuli such as irreparable DNA damage, hypoxia, extremely high cytosolic Ca^2+^ concentrations, and oxidative stress are initiators of the mitochondrial pathway of apoptosis [39]. The key “player” here is the p53 protein, which reaches only low cell concentrations in healthy cells. In stress conditions, p53 is able to detect DNA damage and stop the cell cycle in the G1/S phase. In the absence of repair, the p53 protein stimulates the production of proapoptotic molecules, such as BH3 proteins (BID, BIM, PUMA, Noxa, BIK, BAD, HRK). These proteins activate (in several steps) other proapoptotic proteins resulting in increased permeability of the mitochondrial outer membrane [40,41] and subsequent activation of the intrinsic apoptotic pathway leading to cell death [38,42]. Several chalcones have demonstrated the ability to induce apoptosis by affecting the mitochondrial apoptotic pathway.

Plants of the Glycyrrhiza species belonging to the Fabaceae family have been used in traditional medicine for centuries. The Glycyrrhiza root (licorice root) extract contains many active compounds such as coumarins, saponins, flavonoids, and stilbenoids. Moreover, the licorice root contains also chalcones [43]. Several licorice chalcones have recently shown to have antiproliferative properties by inducing several mechanisms of cell death including apoptosis [44]. Recently, isoliquiritigenin (ISL) has been shown to induce apoptosis in non-small cell lung cancer cells (NSCLC). ISL treatment led to a significant increase in the expression of the proapoptotic protein Bax and caspase-3 activation. On the other hand, the expression of antiapoptotic Bcl-2 -cell protein was decreased. As authors suggested, the proapoptotic effect of ISL may be linked to the modulation of the PI3K/AKT/mTOR signaling pathway [45]. Pro-apoptotic effect of ISL has also been documented in human renal carcinoma Caki cells. Treatment of Caki cells with ISL resulted in caspases-9, -7, -3 activation, PARP cleavage, and cytochrome c release. In addition, ISL increased the expression of the proapoptotic protein Bax and decreased the expression of antiapoptotic proteins Bcl-2 and Bcl-xl [46].

Another chalcone isolated from licorice root is licochalcone A (LicoA). Although the mechanism of antiproliferative effect of LicoA, like in other chalcones, is multifactorial, the proapototic effect of LicoA has been described in several studies. Qiu et al. [47] studied the antiproliferative effect of LicoA in NSCLC. They found a significant decrease of Bcl-xL and Bcl-2 levels with simultaneous increase in caspase-3 activity and cleaved PARP. Similar results were also obtained in breast carcinoma cells where LicoA decreased Bcl-2 levels and induced PARP degradation [48]. Recently, Hong et al. [49] confirmed the ability of this chalcone to induce the mitochondrial pathway of apoptosis in human bladder cancer cells. In this study, the main result showed activation of caspase-9 and caspase-3 with subsequent cleavage of PARP. LicoA also increased Bac/Bcl-2 ratio, induced mitochondrial dysfunction followed by cytochrome c release. Moreover, several other studies also showed the potential of LicoA to induce cell death via mitochondrial pathway of apoptosis [50,51,52].

Although LicoA is probably the most studied compound among licochalcones, other licochalcones including licochalcone B, C, D, and E have also shown proapoptotic effect in several cancer cells. In all of these chalcones, apoptosis had the same features as described in LicoA, i.e., modulation of Bcl-2 family protein levels, release of proapoptotic proteins such as cytochrome c and Smac/Diablo, activation of caspases or PARP cleavage [53,54,55,56].

Well-known xanthohumol (XNT) is a prenylated chalcone found in an extract of hops (*Humulus lupulus* L.), the main source of which is beer brewed. It has a wide range of biological effects including chemopreventive, anti-inflammatory, antiproliferative, and antiangiogenic effects. Xanthohumol-induced apoptosis has been associated with caspase-3 and -9 activation and an increase in Bax/Bcl-2 ratio [57]. Later, Scagliarini et al. [58] observed that treatment of colorectal cancer cells with XNT led to DNA damage with subsequent apoptosis induction. Recently, Liu et al. [59] described a new target for the antiproliferative effect of XNT in colorectal cancer cells—hexokinase 2. Its dysfunction is in correlation with the dysregulation of glucose metabolism in cancer cells. Moreover, hexokinase 2 binds to mitochondrial VDAC and is involved in the maintaining of mitochondria potential. XNT downregulated hexokinase 2, suppressed glycolysis, and promoted cytochrome c release with simultaneous activation of the intrinsic apoptosis pathway. In addition, the proapoptotic effect of XNT can also be associated with Notch signaling pathway, which is implicated in physiological as well as in pathophysiological processes including cell differentiation, proliferation, invasiveness, angiogenesis, metastasis, and apoptosis. In a study of Sun [60], XNT treatment caused suppression of the Notch signaling pathway as well as a decrease in Bcl-2 and Bcl-XL levels, PARP cleavage, and caspase-3 activation. As authors suggest, the antiproliferative and proapoptotic effect of XNT may be related to the regulation of Notch signaling. More details about the anticancer effects of xanthohumol have been recently reviewed by Jiang and co-workers [61].

Isobavachalcone (IBC), another natural chalcone with antiproliferative and antitumor effects, was first found in Psoralea corylifolia (Fabaceae), a plant used in traditional Chinese and Indian medicine [62]. In addition to some of the aforementioned mechanisms, IBC induced apoptosis due to downregulation of Wnt/β-catenin signaling, abnormal activation of which can trigger carcinogenesis [49]. Treatment of colorectal cancer cells with IBC initiated several processes resulting in the mitochondrial apoptosis pathway as documented by changes in Bcl-2/Bax ratio, translocation of Bax to the mitochondria, activation of caspase-3, as well as PARP cleavage. Moreover, the expression of the inhibitors of apoptosis, XIAP and survivin, has been downregulated [63]. Additionally, the propapototic effect of IBC can be also related to the downregulation of ERKs/RSK2 or Akt signaling pathways [64,65,66]. On the other hand, the antiproliferative effect of IBC may also be associated with the induction of non-apoptotic cell death, methuosis-like cell death [67] (see Chapter 3.2.2).

Cardamonin (CAR), a chalcone isolated from plants of the family Zingiberaceae, Asteraceae, Piperaceae, Polygonaceae, and many others, has been studied for many years for its health benefits including anti-inflammatory, antioxidant, and antineoplastic effects [68]. In a study performed in osteosarcoma cells, CAR caused dose-dependent increase in p38 and JNK phosphorylation associated with the inhibition of cell proliferation, reduced migration, and apoptosis induction. Western blot analyses showed increased expression of proapoptotic Bax and BAD proteins and down regulation of antiapoptotic Bcl-2 protein. Moreover, activation of caspase-3 and PARP cleavage has also been observed [69].

Pinostrobin, isolated from Boesenbergia pandurate, has been documented to have a strong antiproliferative and apoptosis-inducing effect on cervical cancer cells. In treated cells, it induced mitochondrial dysfunction and changes in mitochondria structure associated with the release of mitochondrial proteins including cytochrome c, HtrA2/Omi, and Smac/DIABLO. These activate the caspase pathway and finally apoptosis. In addition, proapoptotic proteins of Bcl-2 family including Bad, Bax, have also been overexpressed. At the microscopic level, morphological changes, such as DNA fragmentation, chromatin condensation, cell blebbing, and increased cytoplasmic volume have also been observed [70].

Butein, a chalcone broadly biosynthesized in plants, has been described to possess different biological actions. Recently, an in vitro study has shown that butein induced apoptosis in human cervical cancer cells. In treated cells, butein disturbed mitochondrial transmembrane potential followed by cytochrome c release, caspase activation, and PARP degradation. Furthermore, butein significantly reduced the levels of the antiapoptotic Bcl-xL protein. On the other hand, the levels of the other members of Bcl-2 protein family have not been significantly changed. In addition, proteins of the inhibitor of apoptosis family (XIAP, cIAP-1, and survivin) have been downregulated in butein-treated cancer cells [71]. These authors described a similar mechanism of butein action also in ovarian cancer cell lines [72].

Additionally, the ability to induce the intrinsic pathway of apoptosis has also been described in several natural chacones such as echinatin [73], broussochalcone [74], hydroxysafflor yellow A [75], 3-deoxysappanchalcone [76], sappanchalcone [77], and many others (see also Table 1).

#### 2.1.2. Extrinsic Pathway of Apoptosis

The external apoptotic pathway is initiated by the interaction of death ligands with specific receptors (death receptors). Upon stimulation of the death receptor with a suitable ligand (TNF family), oligomerization and conformational changes occur, leading to the formation of a death-inducing signaling complex (DISC), the function of which is to activate caspase-8. Caspase-8, and in some cases caspase-10, are specific initiating caspases that induce mitochondrial stress and thus trigger the intrinsic apoptosis pathway or directly activate executioner caspases-3, -6, and -7 [38,39].

As it was mentioned, chalcones are phytochemicals with multitargeted activity and several of them are able to induce more than one type of apoptosis.

In above-mentioned study of Jaudan et al. [70], the authors presented the ability of pinostrobin to induce the intrinsic pathway of apoptosis. However, in pinostrobin-treated human cervical cancer cells, they also described characteristic events of the external pathway of apoptosis such as significantly increased expression of TRAIL R1/D4, TRAIL R2/D5, Fas, FADD (Fas associated via death domain). Thus, findings from analyses suggest that pinostrobin-induced apoptosis involves both intrinsic and extrinsic pathways.

Another above-mentioned chalcone, cardamonin, similarly to pinostrobin, can induce both intrinsic and extrinsic apoptosis. In hepatocellular carcinoma cells, it significantly increased the levels of proapoptotic proteins such as FADD, FAS, TRAIL, and increased the activity of caspase-8 and subsequent activation of executioner caspase-3 and -7 [145].

Similarly, licochalcone B (LicoB) has been documented to induce apoptosis via both extrinsic and intrinsic pathways. Kang et al. [97] studied the antiproliferative effect of LicoB using human melanoma and squamous carcinoma cells. LicoB treatment increased the expression of death receptors (DR4 and DR5) as well as increased levels of caspase-8, indicating an extrinsic pathway of apoptosis. In addition, LicoB increased the levels of Apaf-1, Bax, and cleaved-PARP together with increased levels of caspase-9, proteins involved in the intrinsic pathway.

#### 2.1.3. Endoplasmic Reticulum Pathway of Apoptosis

In addition to receptor and mitochondrial-mediated apoptosis, other important targets associated with programmed cell death have been described in recent years.

The endoplasmic reticulum (ER) is an organelle with many important functions such as protein synthesis and folding, calcium buffering, biosynthesis of phospholipids, and carbohydrate metabolism [146]. Disturbances in several homeostatic processes can lead to ER stress response (ER stress). Due to various stress stimuli such as glucose depletion, oxidative stress, viral infections, anticancer drugs (e.g., proteasome inhibitors), etc., unfolded, or misfolded proteins accumulate in the lumen during the folding of proteins from the linear (2D) conformation to the three-dimensional structure (3D) conformation. This leads to the activation of a stress pathway called UPR (unfolded protein response) as an attempt to repair ER stress and restore normal function. As a result of stress in the ER, transmembrane molecules anchored in the ER membrane are released, e.g., PERK protein (PKR-like protein kinase), IRE1 (inositol requiring kinase 1) or ATF6 (activating transcription factor 6) [147]. On the other hand, during prolonged ER stress, UPR executes the cells to apoptosis. Many studies have shown that both intrinsic and extrinsic pathways are involved in ER stress-induced apoptosis [35]. In addition, induction of ER stress has been considered as a potential mechanism for anticancer agents. Furthermore, several experimental works showed that ER stress can play an important role in the cytotoxicity of many phytochemicals [148,149].

The ability of chalcones to induce apoptosis via the ER pathway has also been demonstrated. LicoA was mentioned above as an inducer of the intrinsic pathway of apoptosis [47]. However, these authors in the same article described also the ability of LicoA to induce ER stress. In LicoA-treated human lung cancer cells, they found increased expression of p-EIF2α and ATF, ER stress-related proteins, indicating that ER stress was involved in LicoA-induced cell death.

Recently, Kwak et al. [141] described the proapoptotic effect of echinatin associated with reactive oxygen species (ROS) production and induction of ER stress. They found increased levels of both GRP78 and CHOP, which are markers of ER stress. It is suggested that ROS production and ER stress can disrupt mitochondrial potential with subsequent apoptosis induction. However, like other chalcones, echinatin also modulates the activity of proteins involved in both the extrinsic and intrinsic pathways of apoptosis such as DR4, DR5, proteins of the Bcl-2 family, APAF-1 or PARP suggesting a multifactorial mechanism of echinatin-induced apoptosis.

ER stress and ROS generation are probably also involved in apoptosis in the cell exposed to butein. Butein in lung cancer cells modulated the activity of several proteins of ER stress pathway including PERK, eIFα, ATF4, CHOP, IRE1α, or XBP1 which resulted in the activation of apoptotic machinery. In addition, suppression of SOD2 activity, glutathione depletion, and stimulation of NADPH oxidase activity can also be involved in ROS production in butein-treated cells [150].

Increased ROS production and ER stress pathway play an important role also in flavokawain C-induced apoptosis in colon cancer cells. The induction of apoptosis was associated with the increased expression of GADD153, which is one of the members of the apoptosis mediated by ER stress [151]. The specific mechanisms of chalcone-induced apoptosis are shown in Figure 2.

Although apoptosis is the most studied method of cell death, the antitumor mechanism of chalcones also involves the triggering of non-apoptotic cell death pathways.

### 2.2. Non-Apoptotic Cell Death

Although apoptosis is the most studied type of cell death, other mechanisms have historically been identified, including autophagic cell death and necrosis [152]. In recent years, there have also been discussions about less-known types of non-apoptotic cell death, such as methuosis, ferroptosis, necroptosis, pyroptosis, and others [153,154,155,156,157]. Next, we deal with the mechanism of demise of the cells caused by chalcones.

#### 2.2.1. Autophagic Cell Death

Macroautophagy referred to as autophagy, is in general a prosurvival process. This catabolic pathway provides for the degradation and elimination of cytosolic proteins, macromolecules, and organelles in specific lysosomes. The target structures intended for degradation are enveloped by a double membrane and form the so-called autophagosomes (Figure 3). In them, degradation and proteolytic cleavage of macromolecules and organelles into basic components occur after fusion with lysosomes [158]. Generally, autophagy blocks the induction of apoptosis by inactivation of caspases, and apoptosis-associated caspases activation shuts off the autophagic process by cleavage of autophagy proteins. However, under some circumstances, autophagy also may initiate cell death.

Autophagy is one of the mechanisms that has not yet been fully elucidated, but it is known to play a dual role in tumorigenesis and progression. Autophagic cell death can be effectively used as an alternative in the treatment of tumors that are resistant to apoptosis [159]. One of the best-studied regulators of autophagy is the PI3K/Akt/mTOR signaling pathway, a key cascade that connects mitogenic stimuli from the external environment with the intracellular signaling pathways and plays a key role in controlling cell growth and survival, and the cell cycle. Moreover, it modulates cellular response to apoptotic and proliferative stimuli, regulates cell motility, invasiveness, vascular formation, and inhibits autophagy induction when activated aberrantly [160,161]. mTORC1 controls the process of autophagy by inhibiting the (hyperphosphorylation) complex consisting of Atg1/ULK1, FIP200, Atg13, and Atg101 and plays a crucial role in the induction of autophagy [162]. When activated, this complex triggers the activation of PI3KC3 complex I, thereby producing PI3P and forming a phagophore. After initiation and nucleation of the phagophore, an autophagosome is formed, the formation of which is mediated by other Atg proteins (Atg 9, Atg12, Atg5, Atg7, Atg10, Atg16L, Atg8), LC3 I and II and other macromolecules that ensure a smooth process of autophagosome maturation, fusion with lysosomes, and hydrolytic cleavage of target proteins into amino acids and fatty acids, which are recycled. The complexity of this system can be demonstrated in association with various other signaling pathways such as p53, protein kinase B (Akt), Bcl-2, Ras, UVRAG, Bif-1, and others [161]. Therefore, progress is needed in the study of autophagy and its association with other cellular processes.

Several chalcones have been shown to affect various proteins involved in the autophagic process from its initiation to lysosomal degradation of target structures. Flavokawain B (FKB) isolated from Alpinia pricei Hayata showed multiple effects on melanoma cell lines and human lung adenocarcinoma cells.

Hseu et al. [121] studied the effect of FKB on the induction of autophagic cell death at concentrations of 2.5—10 µg/mL. They observed increased AVO (acidic vesicular organelles) formation in FKB-treated cells. Formation of AVO is a characteristic feature of autophagy. At the molecular level, a dose- and time-dependent decrease in the levels of phosphorylated mTOR, a main negative regulator of autophagy, has been observed. Moreover, Atg4B, a cysteine protease responsible for LC3 delipidation, has also been decreased after treatment with FKB. This decrease may be associated with intracellular accumulation of ROS. In addition, along with an increase in Atg7, there was also an increase in LC3 -I /-II (marker protein for autophagosomes) levels. Atg7 is an E1-like activating enzyme, essential for the successful conversion of LC3-I (cytosolic form) to its conjugated form LC3-II (conjugated LC3 with PE). Furthermore, FKB-induced cell death also “carries” features of mitochondrial apoptosis such as caspase-3 and -9 activation or PARP cleavage, suggesting the involvement of both autophagy and apoptosis in cell death induced by this chalcone.

The same mechanism of cell death was documented by these authors in FKB-treated melanoma cells [122] or in combination with doxorubicin in gastric cancer cells [123].

Recent study revealed that also LicoA induced apoptosis by triggering autophagy in osteosarcoma cells. Data from Western blot analysis showed that the protein level of LC3 -I/-II was significantly increased. In addition, activation of ATM-Chk2 pathway and subsequent G2/M phase arrest has also been observed in LicoA-treated osteosarcoma cells. On the other hand, chloroquine, an autophagy inhibitor, significantly reduced the proapoptotic effect of LicoA [90]. Additionally, the ability of LicoA to induce cell death via autophagy induction has been proved also in melanoma [91], breast [92], or lung cancer cells [93].

In addition, chalcones capable of inducing autophagy include hydroxysafflor yellow A [143], cardamonin [105], panduratin A [163], bavachalcone (isolated from Cullen corylifolium) [111], trans-chalcone [164], or α,2′-dihydroxy-4,4′-dimethoxydihydrochalcone, isolated from Cedrela odorata (Meliaceae) [165].

#### 2.2.2. Other Cell Death Pathways

Although some types of caspase-independent cell death have been less studied until recently, it is appropriate to address them in relation to the possible therapies for cancer. Methuosis, a potentially novel cell death mechanism, independent of caspase activation, is characterized by excessive cytoplasmatic vacuoles (derived from macropinosomes) accumulation in cancer cells exposed to anticancer compounds [35]. There are only a few articles describing methuosis as a cell death mechanism in chalcone-treated cells and most of them were synthetic compounds [166,167]. In above-mentioned paper [67], authors described IBC-induced cell death associated with massive cytoplasmatic vacuolization. It seems that this phenomenon may be useful in cancer cells resistant to apoptosis.

Paraptosis, a type of programmed cell death morphologically distinct from apoptosis and necrosis, has also been demonstrated as a cell death mechanism in chalcone-treated cancer cells. Antiproliferative effect of XNT in leukemia cells was associated with extensive cytoplasmic vacuolization without the involvement of apoptosis or autophagy at a concentration of 15 µM. Morphological changes in cells, changes in the expression of ER stress markers (Chop, Bip/Grp78), and the phosphorylated form of p38 MAPK indicate the induction of paraptosis, which is characterized by the presence of extensive vacuolation originating from the ER and mitochondria [168].

Necroptosis (a programmed form of necrosis), a caspase-independent pathway, is characterized by the modulation of many proteins, including RIPK1 and RIPK3 (receptor-interacting protein kinase 1 and 3) and MLKL (mixed lineage kinase domain-like), which are essential for necrosome formation and death stimulus propagation [169]. Escobar et al. [170] reported the antiproliferative effect of ISL on human neuroblastoma cell line cells. Using a necroptosis inhibitor, necrostatin-1, they found that ISL was likely to induce necroptotic cell death. ISL has caused an increase in the phospho-RIP1/RIP1 ratio and depletion of intracellular ATP levels, which are also markers of necroptosis.

These examples confirm the ability of chalcones to induce different types of cell death, and this opens up a wide range of new possibilities for their use in the therapy of cancer and other diseases.

## 3. Cell Cycle and Tubulins as a Target of Chalcones

The antiproliferative activity of chalcones is closely linked to cell cycle blockade and subsequent induction of apoptosis [171]. Tumor cell transformation, which may be the keystone for the development of cancer, may cause cells to be unresponsive to regulatory mechanisms and begin to divide uncontrollably. In healthy cells, the cell cycle is very strictly regulated. Cell-cycle-regulating proteins include cyclins and cyclin-dependent protein kinases (CdKs), which are activated by CdK-activating kinases (CAKs) upon complex formation. The mechanism of action of chalcones in relation to the cell cycle is pleiotropic and includes an effect on cyclins and cyclin-dependent kinases (CDKs) [47,141,172,173], topoisomerases [113,130], tubulins [131], or CKI expression and degradation [172]. These mechanisms may lead to inhibition of proliferation and induction of apoptosis. Although chalcones can inhibit the cell cycle in different phases [54,57,135], they have been found to act mainly as antimitotic agents inhibiting cell cycle in the G2/M phase.

Shen et al. [90] evaluated the effect of LicoA on the expression of cell cycle regulating proteins in human osteosarcoma cells. The results showed that the levels of Cdc25C, cdc2 (a protein also known as CdK1), its phosphorylated form, and cyclin B1 decreased. Cdc25 is a phosphatase that activates CdK by cleavage of an inhibitory phosphate residue. In response to DNA damage, there has been a significant increase in the activated (phosphorylated) check point kinase Chk2 and ATM kinase, which play a key role in the detection and amplification of genotoxic stress signals and p53 activation. These changes in the level of proteins that regulate the cell cycle have led to G2/M phase arrest in osteosarcoma cell lines. Similar results on lung cancer cells have been documented by Qiu et al. [47]. LicoA inhibited the growth of cancer cells via cell cycle arrest at G2/M, and this effect has been associated with decreased expression of Cyclin B1, Cdc2, and Cdc25C. Another study performed on a glioma cell line showed that LicoA significantly reduced the levels of cyclins A, B1, D1, E1 and CdK 1, CdK2, CdK4, and CdK6 at mRNA as well as protein level. They also noted cell cycle arrest in the G0/G1 and G2/M phases [94].

Echinatin and 2′,4′-dihydroxy-6′-methoxy-3′,5′-dimethylchalcone, isolated from the buds of Cleistocalyx operculatus, act with similar mechanisms in esophageal squamous cell carcinoma cells as well as in multidrug-resistant hepatocellular carcinoma cell line. Proteins that significantly affect the cell cycle include p21 and p27. Increased p27 expression suppresses cdc2 and cyclin B1 expression and regulates cell cycle progression through the G2/M phase (p21 is a promiscuous inhibitor of all cyclin/CdK complexes). Increased GSK3β levels caused downregulation of cyclin D1 [141,173].

Millepachine (MIL), a natural chalcone from Millettia pachycarpa Benth, has been found to suppress cancer cell growth [130]. They found G2/M cell cycle arrest in millepachine-treated hepatocarcinoma cells. Later, to clarify the mechanism underlying the G2/M cell cycle arrest, these authors described the ability of MIL to disrupt the mitotic spindle assembly and to delay the time of microtubules polymerization. On the other hand, the direct effect of MIL on microtubule polymerization has not been observed [174]. Surprisingly, the opposite findings were published by Yang et al. [131], who found that MIL interacts with β-tubulin and binds to the colchicine site.

In addition, several other natural chalcones have been shown to cause G2/M arrest via suppression of cyclin levels or inhibition of CdKs activity including ISL [175], flavokawain A [171], XNT [57], FKB [176], CAR [177], butein [150], and many others.

### Interaction with Tubulins

Microtubules are dynamic cellular structures responsible for various processes in the eukaryotic cell. They play an important role in motility, mitotic apparatus formation and cell division, angiogenesis, metastasis, maintenance of cell shape, and transport of organelles and other cell components [178]. The major component of microtubules is the heterodimeric globular protein tubulin. Dimers of α and β subunits with a molecular weight of approximately 55 kDa form a regular tubular system. In terms of the action of conventional cytostatics, the most common mechanism of action is inhibition of tubulin polymerization or inhibition of microtubule depolymerization, which lead to cell cycle arrest in the G2/M phase and subsequent apoptosis.

Recently, Liu et al. [179] studied antiproliferative effect of six chalcones isolated from the seeds of Cullen corylifolium including 4′-*O*-methylbroussochalcone B, broussochalcone B, isobavachalcone, bavachromene, isobavachromene, and dorsmanin A. All of them suppressed the growth of the different leukemic cell lines but only 4′-*O*-methylbroussochalcone B exhibited its effect by inhibition of tubulin polymerization with subsequent blocking of the cell cycle at the G2/M phase. Another chalcone, flavokawain A, similarly inhibited tubulin polymerization as well as the cell cycle at the G2/M phase, and this effect was associated with apoptosis induction in prostate cancer cells [124].

## 4. Inhibition of Topoisomerases

DNA topoisomerases are nuclear enzymes that are essential for the proper progression of replication, and therefore cells are extremely sensitive to the loss of their function. Topoisomerases, via a mechanism mediated by ATP, bind DNA, cause the break, release excessive torsion, rejoin the broken strand, and allow the replication process to continue. Topoisomerase I (TOPO I) relaxes supercoiled DNA, and topoisomerase II (TOPO II) allows chromosome separation in mitosis [180]. Human topoisomerase II α reaches higher levels in the S and G2/M phases, topoisomerase II β does not change its concentration during the cell cycle and is overexpressed in proliferating cancer cells. From a molecular viewpoint, there are two important targets for drugs interacting with topoisomerase II, the ATPase domain, and the central DNA-binding core [181]. Topoisomerase II inhibitors also have an intercalating effect and can cause a wide range of chromosomal aberrations. They induce the stabilization of Topo II-DNA complexes, which are easily cleaved or disrupt the catalytic activity of the enzyme, leading to damage of the double-stranded DNA. DNA breaks (single-stranded and/or double-stranded) caused by inhibition of topoisomerases induce DNA repair or apoptosis mediated by p21 and/or p53 proteins [182].

In the group of naturally occurring chalcones, MIL acts as a TOPO II inhibitor, inhibiting topoisomerase II activity. This effect was associated with DNA double strand breaks and a concentration-dependent increase in the levels of γ-H2AX protein, a phosphorylated form of H2AX, which is considered as a marker of DNA damage [130]. On the other hand, isoliquiritigenin has been found to inhibit TOPO I. ISL-treated glioma cells underwent apoptosis and growth inhibition, and this effect has been associated with a strong, reversible inhibitory effect on TOPO activity. Moreover, ISL was as potent as camptothecin, a clinically used anticancer drug [183]. In addition, a recent study reported that isoliquiritigenin and echinatin isolated from the roots of Isodon ternifolius (Lamiaceae) act as dual inhibitors of Topo I and TDP1. TPD1 (tyrosyl-DNA phosphodiesterase 1) is a DNA-repair enzyme that catalyzes topoisomerase I-induced DNA repair and therefore may potentiate the therapeutic effect of TOPO I inhibitors. This hypothesis was confirmed because both chalcones show a synergistic effect with topotecan in breast cancer cells [184]. Ability to inhibit either TOPO I or TOPO II has also been documented for pinostrobin [113] or quinochalcone [185].

## 5. Inhibition of p53 Regulators

The p53 protein, the “guardian of the genome”, is a key tumor suppressor regulating target genes involved in cellular processes such as DNA repair, cell cycle regulation, initiation of apoptosis and autophagy, regulation of cellular metabolism, senescence, and others. Loss of its function, whether by mutation at the gene level, or by inactivation by means of stabilization and degradation, usually leads to the initiation and progression of carcinogenesis [186]. Low levels of p53 in cells are maintained by various mechanisms including post-translational modifications, phosphorylation, ubiquitination, and others. Increase of the expression of p53 negative regulators (especially ubiquitin ligases and associated enzymes) leads to its proteasome degradation [187]. Due to the important role of p53 in regulation of the growth and survival of cells, it is a suitable target for the study of natural substances with antiproliferative and antitumor effects.

### 5.1. Inhibition of MDM2 and Other Proteins

In addition to mutations in the p53 gene, which have been found in approximately 50% of human cancers, overexpression of MDM2 and MDMX is another mechanism leading to the suppression of p53 function. Mouse double minute 2 (MDM2) is the main negative physiological regulator of p53 expression, which, by stimulating ubiquitin-proteasome degradation of p53, keeps its expression at a low level. In cells exposed to genotoxic stress, MDM2 catalyzes the mono-ubiquitinization of p53, resulting in its export from the nucleus to the cytosol. Another protein essential for the inhibition of p53 function is MDMX, a homologue of MDM2 that is also overexpressed in many tumor types, thus disruption of the three-protein complex—MDMX/MDM2/p53—offers another option for the pharmacological treatment of wild-type p53 tumors [187]. Natural chalcone butein has been shown to have antiproliferative effects by reducing the levels of phospho-MDM2 and the other key proteins involved in cell proliferation. Woo et al. [101] documented p53-dependent apoptosis in butein-treated chronic myeloid leukemia cells. This effect was associated with the degradation of MDM2. Later, it was demonstrated that butein blocked the interaction between MDM2 and p53, resulting in the suppression of MDM2-mediated p53 ubiquitination [188]. Similar effect on p53 pathway i.e., increased expression of p53 and decreased expression of MDM2 has also been found in other natural chalcones including ISL [46], LicoA [47], XNT [104], broussochalcone [74], and cardamonin [189].

### 5.2. Inhibition of p53 Deacetylases

The p53 study showed that in addition to inhibiting its degradation, post-transcriptional and post-translational modifications, which include acetylation and deacetylation, also significantly affect its function (Figure 4). Acetylation directly affects the activity of this key protein, so the loss of acetylated sites inhibits its ability to regulate the cell cycle and induces cell death [190]. HDACs affect cell transcription not only by deacetylation of histones, but they also regulate deacetylation of non-histone proteins such as p53, STAT3, Hsp90, Akt, NF-κB, and others [191]. Natural deacetylating enzyme inhibitors such as HDAC1 or SIRT1 increase the levels of acetylated p53 and thus promote its tumor-suppressive antitumor activity [190]. Analogs of the bichalcones rhuschalcone IV and rhuschalcone I obtained from the plant Rhus pyroides (Anacardiaceae) in both silico and in vitro screening showed activity against SIRT1 and 2 [192]. Naturally occurring rubone also potentiated the antitumor activity of paclitaxel (PTX) in PTX-resistant prostate tumor cells DU145-TXR and PC3-TXR. The combination of rubone (5 and 10 µM) and PTX significantly increased the expression of miR-34a, a tumor suppressor that is downregulated in resistant tumor cells. Increased miR-34a expression led to increased p53 levels and decreased SIRT1 levels [117].

## 6. Antiangiogenic Effects of Chalcones

The blood supply of the tumor is one of the critical factors determining its growth, progression, and metastasis. The formation of new blood vessels depends on many pro- and anti-angiogenic factors and their related receptors. The main pro- angiogenic factors include VEGF (vascular endothelial growth factor), the expression of which is regulated by hypoxia. It stimulates endothelial cell proliferation and migration, vascular permeability, and expression of matrix metalloproteinases (MMPs). Other important factors are FGF-2 (fibroblast growth factor 2), PDGF (platelet derived growth factor), TGF-β (transforming growth factor-beta), angiopoietins, ephrins, apelin (APLN), and chemokines [193].

The effect of chalcones on tumor angiogenesis was recently described in a detailed review [194], so there are just a few of the latest findings from recent years below.

Wang and co-workers [78] demonstrated inhibition of angiogenesis in zebrafish embryo angiogenesis as well as in rabbit corneal neovascularization models with ISL. In glioma cells, they found decreased production of proangiogenic factors including VEGF, FGF-2, and TGF-β with Akt downregulation. These effects were partly reversed by prostaglandin E2. As authors suggest, chalcone ISL inhibits angiogenesis, at least partly through the inhibition of COX-2 signaling.

Xanthohumol, another chalcone studied as a potential antiangiogenic compound, has been shown to inhibit the secretion of VEGF and IL-8 in pancreatic cancer cells due to suppression of NF-κB activity. In addition, co-cultivation of pancreatic cancer cells with human umbilical vein endothelial cells (HUVECs) enhanced tube formation in HUVECs. Xanthohumol, even in nanomolar concentrations, significantly inhibited the formation of capillary-like tubes by HUVECs [195]. Moreover, XNT, except its proapoptotic effect, has been found to reduce the production of VEGF in multiple myeloma cells, indicating that XNT may also suppress cancer-cell-stimulated angiogenesis [104]. Suppression of VEGF release has also been documented in butein-treated melanoma cells and this effect can be a result of the inhibition of PI3K/Akt/mTOR signaling pathway [196].

Anticancer effect of flavokawain B can also be attributed to its antiangiogenic effect. FKB in nontoxic concentrations inhibited endothelial cell proliferation, migration, and tube formation. Antiangiogenic effect of this chalcone was also confirmed in in vivo conditions in a zebrafish model [197].

Hypoxia is a key stimulus that induces angiogenesis via the activation of hypoxia-inducible transcription factor (HIF-1) with subsequent activation of other proangiogenic factors [198]. Recently, cardamonin has been reported to block the expression of HIF-1α at mRNA and protein levels in a triple negative breast cancer cell line. This effect was associated with the repression of the mTOR/p70S6K pathway and increased ROS production [106].

One of the important regulators of angiogenesis is matrix metalloprotinases (MMPs), of which MMP-2, -9, and -14 are mostly involved in tumor angiogenesis [199]. Isoliquiritigenin has been reported to inhibit stimulators of angiogenesis including VEGF, FGF-2, and TGF-β [78]. However, it seems that the antiangiogenic activity of ISL is more complex and may also include the ability to inhibit MMPs. Zhang et al. [200] found that ISL downregulated the expression of several MMPs including MMP-9 at mRNA as well as protein level. Additionally, suppression of MMP-9 expression has also been reported in butein-treated oral squamous cell carcinoma [102].

Isoliquiritin apioside (ISLA), another natural chalcone found in the licorice root, was found to inhibit some steps in angiogenesis. Under hypoxic conditions, this chalcone significantly decreased the level of nuclear HIF-1α in a dose-dependent manner. Furthermore, ISLA reduced the levels of MMP-9 and placental growth factor (PlGF) under both normoxic and hypoxic conditions. The levels of VEGF were only slightly decreased in ISLA-treated human fibrosarcoma cells. In addition, ISLA effectively blocked the migration of HUVECs as well as capillary tube formation of HUVECs. This antiangiogenic effect has also been confirmed in CAM assay where ISLA significantly suppressed VEGF-induced angiogenesis [201].

## 7. Modulation of Selected Signaling Pathways

Regulation of some signaling pathways may be of great importance in the treatment of cancer, as it has been found that the signaling of these cascades is often genetically altered in many tumors. Their activity results in a cellular response such as changes in the transcription and translation of target proteins, cell proliferation, differentiation and growth, migration, invasiveness, and metastasis. Key cascades include the RTK/RAS pathway, which is deregulated in up to 46% of tumors. Other examples are PI3K, Nrf2, Wnt, Notch, NF-κB, Jak/STAT, Hippo-YAP, and the others [202,203].

PI3K/Akt/mTOR pathway, crucial for the proliferation, growth, and survival of tumor cells, is often overactivated in various cancers and it appears to be a potential target for anticancer therapy. Until today, although several PI3K/Akt/mTOR inhibitors have been studied in clinical trials in cancer patients, only a few of them (e.g., temsirolimus, everolimus, copanlisib) have been approved for clinical use in the treatment of different cancers [204]. In addition, several natural agents, including chalcones, have been tested as potential inhibitors of PI3K/Akt/mTOR pathway in tumors.

Many chalcones have been shown to be effective modulators of this pathway, so here we only give examples of natural chalcones not mentioned above. Natural 4-hydroxyderricin and xanthoangelol, in addition to inducing apoptosis in human melanoma cells, suppressed their growth by targeting BRAF and PI3K pathways. Moreover, treatment of melanoma cells with both chalcones led to a significant inhibition of phosphorylated Akt, downstream of PI3K in a dose-dependent manner [119].

Calomelanone, a dihydrochalcone isolated from Cyathostemma argenteum, has been found to be toxic toward different cancer cells. It was found to induce an intrinsic pathway of apoptosis in cancer cells. Calomelanone also induced the activation of caspase-3, -8, and -9 and modulated the expression of proteins of Bcl-2 family. Moreover, the process of autophagy has also been initiated in calomelanone-treated cells. These effects were associated with decreased expression of Akt [205]. Recently, Song and co-workers [206] studied antiproliferative effect of ISL in hepatocellular carcinoma. Similarly, to calomelanone, ISL induced death of cancer cells via initiation of both apoptosis and autophagy. It is suggested that cell death machinery has been associated with modulation of PI3K/Akt/mTOR pathway as ISL suppressed the phosphorylation of Akt, PI3K, and mTOR in cancer cells. Ability of ISL to induce apoptosis via reduction of Akt and mTOR phosphorylation has also been shown in lymphoma cells [207]. In addition, also LicoA induced inhibition of cancer cell growth as well as inhibited cancer cell migration and invasion. These effects were associated with significant inhibition of Akt phosphorylation and PI3K expression in oral squamous cell carcinoma [208].

Wnt/β-Catenin pathway plays a key role in maintaining cellular homeostasis. It regulates a broad spectrum of biological processes including cell differentiation and proliferation, embryo development, or apoptosis. Furthermore, abnormal activation of the Wnt/β-Catenin pathway has been shown in different cancers including colorectal, lung, liver, or breast cancer [209]. Additionally, experiments have also demonstrated that suppression of the Wnt pathway inhibits the proliferation of cancer cells. Therefore, many drugs as potential blockers of Wnt/β-Catenin pathway have entered either preclinical or clinical studies [210]. However, none of them has been approved for clinical use yet.

On the other hand, several plant constituents have been shown as Wnt/β-catenin inhibitors. Derricin and derricidin, two little-known chalcones isolated from Lonchocarpus sericeus, have been investigated as potential anticancer compounds. Both chalcones in colorectal cancer cells significantly reduced the nuclear localization of β-catenin. Moreover, they also inhibited Wnt-reporter activation and these results showed that both chalcones were capable of inhibiting the Wnt/β-catenin pathway in the colon cancer cell line. In addition, this effect was associated with reduced cell proliferation and accumulation of cells in the G2/M phase of the cell cycle [134]. Similar effect has also been observed in lonchocarpin, another chalcone isolated from Lonchocarpus sericeus. This chalcone suppressed the migration and proliferation of colorectal cancer cells. Detailed study showed that the antiproliferative effect has been closely associated with the ability of lonchocarpin to inhibit Wnt/β-catenin pathway in lonchorapin-treated cancer cells. It was found that lonchocarpin decreased nuclear level of β-catenin and prevented its interaction with other proteins or DNA. In addition, inhibition of Wnt/β-catenin signaling has also been confirmed in in vivo experiments using Xenopus laevis embryo model [128].

Cardamonin has been above mentioned as an inducer of either apoptotic or non-apoptotic cancer cell death. In addition, CAR has also been shown as an inhibitor of Wnt/β-catenin pathway in breast cancer cells. Shrivastava et al. [211] documented reduced nuclear translocation of β-catenin as well as inhibition of Wnt-induced signaling in CAR-treated cancer cells. Suppression of Wnt/β-catenin pathway was associated with depressed invasion and migration of cancer cells and inhibition of EMT.

Brousschalcone has also been shown to modulate Wnt/β-catenin pathway function. Shin et al. [142] found that this chalcone promoted the phosphorylation of β-catenin and reduced its levels by increasing its ubiquitination and degradation in the proteasome. Another feature of this chalcone action was the downregulation of the expression of β-catenin-dependent genes such as cyclin D1, c-Myc, axin2 and the activation of apoptosis-mediated proteins such as caspase 3/7 in liver and colon cancer cells.

Notch signaling is necessary for cell-to-cell communication and is involved in a broad spectrum of biological processes including differentiation, proliferation, and survival [212]. On the other hand, abnormal activation of Notch signaling can be associated with malignant transformation via stimulation of proliferation, restriction of differentiation, or avoiding apoptosis in tumor cells. Currently, there are several inhibitors of Notch signaling either in preclinical or clinical investigations [213].

Moreover, some chalcones have also been presented as Notch signaling inhibitors. Xanthohumol was described as a powerful antiproliferative compound. Kunnimalaiyaan et al. [214] studied the molecular mechanism of its action in hepatocellular carcinoma. Among other things, XNT inhibited Notch signaling via inhibition of the expression of Notch1 with subsequent depletion of HES-1, cyclin D1, and survivin proteins, its downstream targets. As authors suggested, this mechanism has been responsible for the suppression of cancer cells growth and initiation of apoptosis.

Later, Walden et al. [215] documented Notch inhibitory effect of XNT also in cholangiocarcinoma. The suppression of cholangiocarcinoma growth has been associated with the reduced expression of Notch1 and Akt in a concentration-dependent manner. Notch signaling is inhibited by XNT also in pancreatic cancer [216] or in doxorubicin-resistant breast cancer cells [217].

Isoliquiritigenin is another natural chalcone able to inhibit cancer proliferation via modulation of Notch signaling. Lin et al. [79] observed that ISL inhibited the proliferation of human glioma stem cells and this effect can at least partially be mediated by the downregulation of Notch1 and its downstream target HES1 at the protein as well as mRNA levels.

Notch-blocking activity has also been shown in butein. Exposure of leukemic cells to butein led to a concentration-dependent suppression of Notch signaling activity as documented by decreased levels of the intracellular domain of Notch1 and Notch3 receptors [218].

The JAK/STAT pathway is involved in the regulation of several biological processes as a key signaling mechanism for a multitude of cytokines and growth factors. On the other hand, dysregulation of this signaling pathway plays an important role in cancer progression [219]. To date, several JAK/STAT inhibitors are investigated in clinical studies in cancer patients [220]. In addition, numerous phytochemicals, including chalcones, have been shown to inhibit JAK/STAT signaling in cancer cells.

In 2016, Dokduang et al. [221] studied the effect of xantohumol on the growth and proliferation of human cholangiocarcinoma cells. They found that XNT in dose-dependent manner inhibited IL-6-induced STAT3 activation followed by the growth inhibition of cancer cells and apoptosis induction. Similar effect of XNT has also been documented by Jiang et al. in human pancreatic cancer cells [222].

Recent work of Song et al. [98] revealed the ability of licochalcone B to inhibit JAK/STAT signaling. In esophageal squamous cell carcinoma cells, LicoB decreased cell growth and induced apoptosis. These effects were associated with blocking JAK2 activity, inhibition of STAT3 phosphorylation, and decreased expression of STAT target protein Mcl-1.

Additionally, other lichochalcones including licochalcone H, C, and D have also been described as inhibitors of JAK/STAT signaling in oral squamous cell carcinoma [223,224,225].

Isoliquiritigenin has been mentioned several times as a chalcone with strong antiproliferative effect in different cancer cells and with different mechanisms of action. In addition to the above mentioned, ISL is also a regulator of the JAK2/STAT3 pathway. In renal carcinoma cells, ISL induced apoptosis. Except “classical” mechanisms such as caspase activation, PARP cleavage, or modulation of phosphorylation of Bcl-2 family proteins, ISL also inhibited the phosphorylation of STAT3 with subsequent reduction of cyclin D1 and D2. Moreover, ISL also inhibited phosphorylation of JAK2, a STAT3 upstream kinase [46]. The ability of ISL to decrease the phosphorylation of STAT3 has also been confirmed later by Wang et al. [226] in human hepatocellular carcinoma cells.

It is suggested that cardamonin-induced apoptosis can also be associated with the inhibition of STAT3 activation. Wu et al. [14] found that in glioblastoma stem cells, CAR inhibited the phosphorylation of STAT3, blocked its nuclear translocation, and attenuated the expression of VEGF, survivin, Bcl-XL, Bcl-2, a downstream genes of STAT3. Similar effect of CAR on STAT3 signaling, i.e., suppressed phosphorylation of STAT3 and blockade of nuclear translocation, has also been described in prostate cancer cells [107].

NF-κB pathway plays an important role in the regulation of the expression of different proteins involved in the immune and inflammatory responses. Moreover, numerous genes involved in apoptosis and cell survival are also under the control of NF-κB pathway. It has been found that constitutive activation of this pathway promotes cell proliferation and survival, tumor invasion, or protection against apoptosis in cancer cells [227].

Butein, first isolated from the plant Oxicodendron vernicifluum, has been shown in a human oral squamous cancer cell line as an inhibitor of the phosphorylated form of NF-κB (p-p65). Moreover, butein also downregulated several NF-κB-regulated proteins such as the antiapoptotic factor survivin, COX-2, and adhesion molecule MMP-9 [102].

Another natural chalcone, isoliquiritigenin, has been mentioned several times as a potent apoptosis inductor or angiogenesis inhibitor. One of the possible mechanisms of its action can be the modulation of NF-κB pathway. Recently, Wang et al. [226] studied the antiproliferative effect of ISL in human hepatocellular carcinoma cells. Among other processes, ISL regulated NF-κB signaling via potentiation of the inhibitory effect of IκB (a natural inhibitor of NF-κB) as well as via downregulation of NF-κB expression. These effects were associated with mitochondrial apoptosis, cell cycle arrest, and ROS production.

In addition, several other signaling pathways were identified as a target of natural chalcones, including ERK1/2 signaling [64,228,229], MAPK pathway [230,231], or FOXO3a pathway [232] (Figure 5).

## 8. Modulation of Other Molecular Targets

Special attention should be paid to multidrug resistance (MDR), which is characterized by the rapid expulsion of cytostatics from the cell by means of transport proteins from the ABC family (ATP-binding cassette) and thus reduces or abolishes the effect of substances used as chemotherapeutics. The ability to inhibit the expression and/or activity of these has been studied with several chalcones. Of the natural ones, Lico A suppressed the MDR-1 expression at the mRNA and protein levels in the triple negative breast cancer cell line [233]. Recently, Wu et al. [234] discovered that LicoA reversed MDR mediated by ABCG2, a member of the ABC transporter family. Licochalcone A suppressed ABCG2-mediated resistance and restored the chemosensitivity of cancer cells to mitoxantrone and topotecan, a well-known ABCG2 substrates. Subsequently, LicoA significantly potentiated topotecan-induced apoptosis in cancer cells. Furthermore, it has been shown that also flavokawain A downregulated p-glycoprotein (MDR1/ABCB1) expression via inhibition of the PI3K/Akt signaling pathway in paclitaxel (PTX)-resistant lung cancer cells in a dose- and time-dependent manner [235]. The ABC protein family also includes MRP1 (ABCC1) and MRP2 (ABCC2) proteins, the level of which was statistically significantly downregulated by millepachine in cisplatin-resistant human ovarian cancer cells, known for the upregulation of these proteins [132].

Ability to modulate the activity of proteins involved in MDR has also been shown in xanthohumol [236], isobavachalcone [237], or isoliquiritigenin [238].

## 9. Conclusions

Chalcones have been intensively investigated as potential anticancer compounds in the past decades. The reason of this review was to collect the knowledge on molecular mechanisms of antiproliferative action of chalcones and draw attention to this group of phytochemicals that may be of importance in the treatment of cancer disease.

In the present paper, we have presented pleiotropic pharmacological effects of natural chalcones. As antiproliferative agents, they interfere with the processes such as inhibition of cancer cells growth or cell death induction. Additionally, their antiproliferative effect may also be mediated by cell cycle arrest, stabilization of p53 or modulation of various signaling pathways associated with cell survival or death. Some studies also reported antiangiogenic effect of chalcones suggesting their impact on tumor environment.

Cancer is a complex disease and it is generally accepted that simultaneous modulation of different targets might be associated with better therapeutic response. In this context chalcones could emerge as a perspective compounds for the development of an anticancer drug with multitarget action. However, despite the results achieved in this area of research, further experimental and clinical investigations are required.

## Figures and Tables

**Figure 1 cancers-13-02730-f001:**
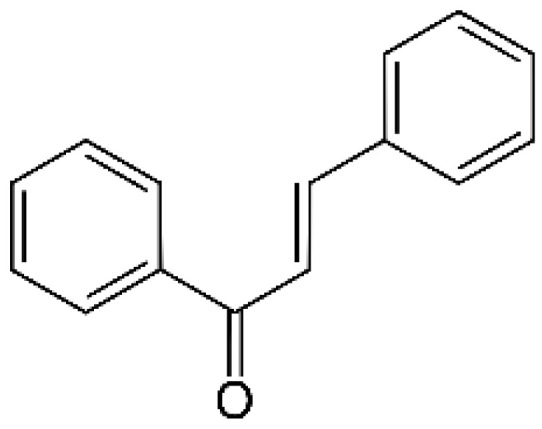
Basic chalcone structure. Original figure made for this review using the Zoner Callisto 5 software.

**Figure 2 cancers-13-02730-f002:**
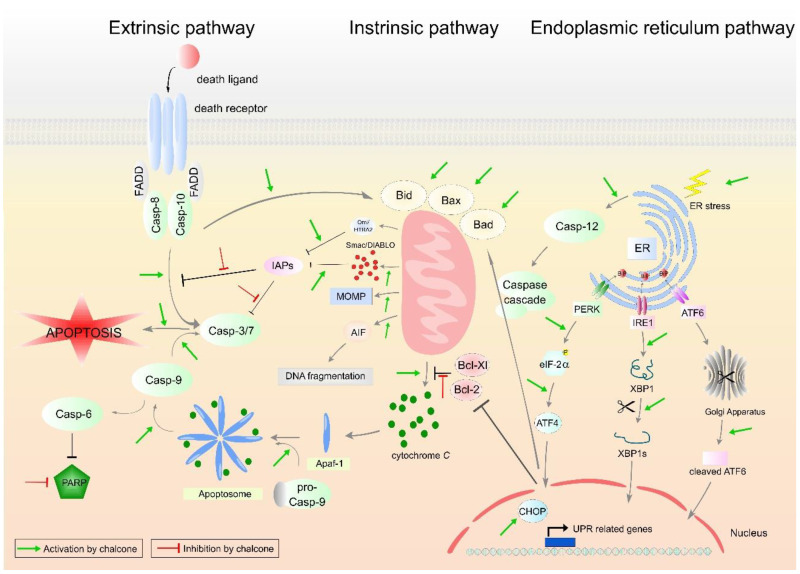
Molecular mechanism of extrinsic, intrinsic and endoplasmic reticulum pathways involved in apoptosis. Abbreviations: (pro)Casp—(pro)caspase, AIF—apoptosis inducing factor, APAF-1—apoptotic protease activating factor 1, ATF 4/6—activating transcription factors 4/6, BiP—binding immunoglobulin protein, Bcl-2/Bcl-Xl—antiapoptotic factors, Bad/Bax/Bid—proapoptotic factors, eIF-2α—eukaryotic translation initiation factor 2 A, ER—endoplasmic reticulum, FADD—Fas associated via death domain, CHOP—C/EBP homologous protein, IAPs—inhibitors of apoptosis, IRE1—inositol-requiring enzyme 1, MOMP—mitochondrial outer membrane permeabilization, Omi/HTRA2—Omi/high temperature requirement factor A2, PARP—poly (ADP-ribose) polymerase, PERK—PKR-like endoplasmic reticulum kinase, Smac/DIABLO—second mitochondria-derived activator of caspase/direct inhibitor of apoptosis-binding protein with low pI, UPR—unfolded protein response, XBP1—X-box binding protein 1. Original figure made for this review using the Zoner Callisto 5 software.

**Figure 3 cancers-13-02730-f003:**
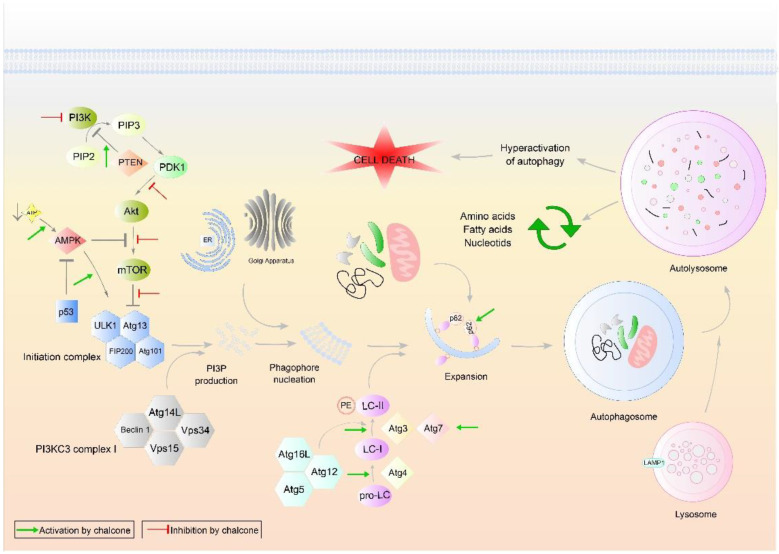
The main steps of autophagy in cells. Abbreviations: (pro)LC I/II—(pro)microtubule-associated proteins 1A/1B light chain 3B, Akt—protein kinase B, AMPK—AMP-activated protein kinase, Atg 3/4/5/7/12/13/14L/16L/101—autophagy-related proteins, ATP—adenosine triphosphate, ER—endoplasmic reticulum, FIP200—family-interacting protein of 200 kD, LAMP1—lysosomal-associated membrane protein 1, mTOR—mammalian target of rapamycin, p53—p53 protein, p62—sequestosome 1, PDK1—phosphoinositide-dependent kinase-1, PE—phosphatidylethanolamine, PI3K—phosphoinositide-3-kinase, PI3KC3 complex I—phosphatidylinositol 3-kinase catalytic subunit type 3 complex I, PIP2—phosphatidylinositol-2-phosphate, PIP3—phosphatidylinositol-3-phosphate, PTEN—phosphatase and tensin homolog, ULK1—Unc-51 like autophagy activating kinase, Vps 15/34—vacuolar protein sorting proteins. Original figure made for this review using the Zoner Callisto 5 software.

**Figure 4 cancers-13-02730-f004:**
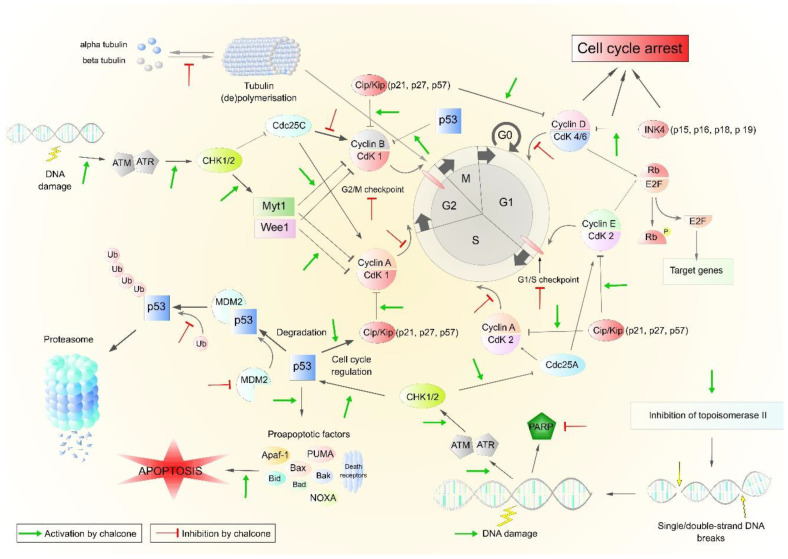
Basic steps in cell cycle regulation and p53 function. Abbreviations: Apaf-1—apoptotic protease activating factor 1, ATM—ataxia-telangiectasia mutated, ATR—ataxia telangiectasia and Rad3 related, Bad/Bak/Bax/Bid—proapoptotic factors, Cdc25 A/C—cell division cycle 25 homolog A/C phosphatase, CDK 1/2/4/6—cyclin-dependent kinases, Cip/Kip—CDK interacting protein/Kinase inhibitory protein, E2F—E2 factor, CHK 1/2—checkpoint kinases, INK4—inhibitors of CDK4, MDM2—Mouse double minute 2 homolog, Myt1—myelin transcription factor 1, Noxa—phorbol-12-myristate-13-acetate-induced protein 1, p53—protein 53, PARP—Poly (ADP-ribose) polymerase, PUMA—p53 upregulated modulator of apoptosis, Rb—retinoblastoma protein, Wee1—inhibitor of CDK1. Original figure made for this review using the Zoner Callisto 5 software.

**Figure 5 cancers-13-02730-f005:**
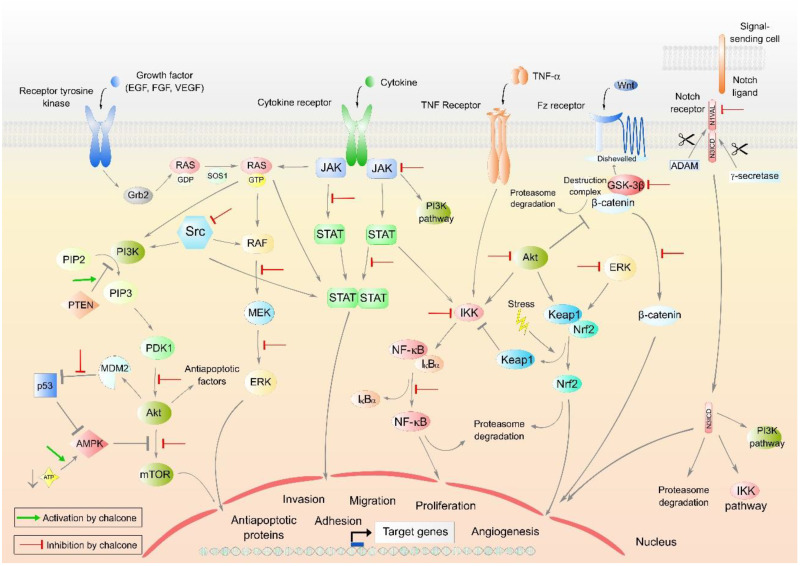
Major oncogenic intracellular signaling pathways. Abbreviations: ADAM—a disintegrin and metalloprotease protein, Akt—protein kinase B, AMPK—AMP-activated protein kinase, ATP—adenosine triphosphate, ERK—extracellular signal-regulated kinase, Fz receptor—frizzled receptor, GDP—guanosine diphosphate, Grb2—growth factor receptor-bound protein 2, GSK-3 β—glycogen synthase kinase-3 β, GTP—guanosine triphosphate, IKK—IκB kinase, IκBα—inhibitor of nuclear factor kappa B, JAK—Janus kinase, Keap1—Kelch-like ECH-associated protein 1, MDM2—mouse double minute 2 homolog, MEK—MAPK/Erk kinase, mTOR—mammalian target of rapamycin, N1VAL—activated Notch1 receptor domain, N3ICD—intracellular domain of Notch3, Nf-Kb—nuclear factor kappa B, Nrf2—nuclear factor erythroid 2-related factor 2, p53—protein 53, PDK1—phosphoinositide-dependent kinase-1, PI3K—phosphoinositide-3-kinase, PIP2—phosphatidylinositol-2-phosphate, PIP3—phosphatidylinositol-3-phosphate, PTEN—phosphatase and tensin homolog, RAF—rapidly accelerated fibrosarcoma, RAS—rat sarcoma protein, SOS 1—SOS Ras/Rac guanine nucleotide exchange factor 1, Src—proto-oncogene tyrosine-protein kinase Src, STAT—signal transducer and activator of transcription protein, TNF α—tumor necrosis factor α, Wnt—wingless-related integration site protein. Original figure made for this review using the Zoner Callisto 5 software.

**Table 1 cancers-13-02730-t001:** Molecular targets of selected natural chalcones in cancer cells.

Chalcone	Structure	Mechanism of Action	Reference
Isoliquiritigenin	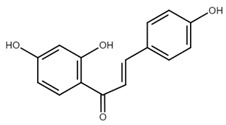	sub-G1, G2/M phase arrest ↑ ROS, p53, p21, p27, cleaved form of caspase-9, -7, -3 and PARP, Bax, Bim, cytochrome c release, LC3 -I/-II, Beclin 1 ↓Mdm2, Bcl-2, Bcl-xl, STAT3, cyclins D1, D2, MMP-2, MMP-9, phospho-AKT, phospho-mTOR, phosphor ERK ½, stability of HIF-1α, p62/SQSTM1, E-cadherin, VEGF, FGF-2, TGF-β, expression of EGFR	[44,45,46,78,79,80,81,82,83,84,85,86,87,88,89]
Licochalcone A	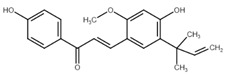	G0/G1, G2/M phase arrest ↑ cleaved form of caspase-10, -8, -3, PARP, activation ATM and Chk2, LC3 -I/-II, Wee1, p21, expression of DR3, DR5, Fas, Bad, Bax, Bak, PUMA, phospho-PERK, ATF4, p-EIF2α, ROS ↓survivin, cyclin B1, CdK1, Cdc2, Cdc25, MDM2, PI3K, Akt, mTOR, PKCε, Sp1p70S6K	[47,49,50,90,91,92,93,94,95,96]
Licochalcone B	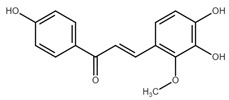	Increase sub-G1 phase ↑ cytochrome c release, p21, p27, levels of caspase-3, -7, -9, cleaved PARP, Bax, p53, CHOP, DR4, DR5, Apaf-1 ↓ MMP, Cyclin A, CdK2, Cdc25 A, Bcl-2, Bid, survivin, Mcl-1, Bcl-xl, Cyclin B1, phospho-JAK2, phospho-STAT3, Mcl-1	[54,97,98,99,100]
Butein	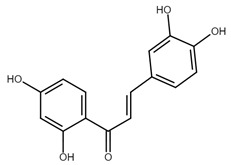	Inhibition of HDAC ↑ phospho-p53, p21, cleaved caspase-9, -8, -3, cleaved PARP, cytochrome c release, Bax, Bad ↓ CdK1, CdK2, CdK4, Cyclin A, B, D, E, phospho-MDM-2, Bcl-xl, Bcl-2, XIAP, survivin, cIAP -1, -2, MMP-9, COX-2, NF-κB, phospho-NF-κB	[72,101,102,103]
Xanthohumol	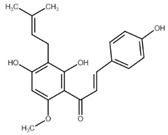	G1/S phase arrest ↑ p21, p53, phospho-p53, phospho-γH2AX, phospho-ATR, phospho-ATM, activation of caspase-3, -9, Bax, cleaved caspase-8, -9, ↓ Cyclin A, B, D, E, CdK1, CdK2, CdK4, Bcl-2, Bcl-xl, Notch1, Ki-67, survivin, Hes1, Hey1	[57,58,60,104]
Cardamonin	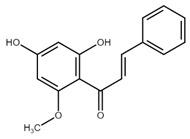	Inhibition of migration, G2/M phase arrest ↑ cleaved caspase-3, -8, -9, PARP, Bax, Cytochrome c, E-cadherin, LC3 -I/-II, Atg5, Beclin 1 ↓ Bcl-2, Mcl-1, Bcl-xl, phospho-NF-KB (p-p65), p-IKK α/β, phospho-PI3K, phospho-Akt, phospho-mTOR, phospho-P70S6K, N-cadherin, ZEB1, Cyclin D1, E, CdK2, CdK4, Ki-67, phospho-JAK2, STAT3, phospho-STAT3, VEGF, MMP-9, COX-2, XIAP, HIF-1α, survivin	[68,105,106,107,108,109,110]
Panduratin A	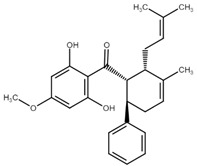	G0/G1 phase arrest ↑ p21, p27, Bax, cytochrome c, cleaved PARP, LC3B -I/II, Atg12, phospho-AMPK ↓ CdK4, Cyclin D1, Bcl-2, p62/SQSTM1	[111,112]
Pinostrobin	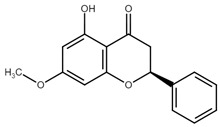	G1/S phase arrest, Inhibition of topoisomerase I ↑ ROS, Bad, Bax, cleaved caspase-3, cytochrome c, TRAIL R1/DR4, TRAIL R2/DR5, FADD, Fas, HTRA2/Omi, p21, SMAC/Diablo, TNF R1 ↓ GSH, NO2-, MMP	[70,113]
Garcinol, isogarcinol	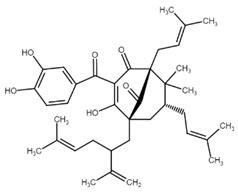	Inhibition of colony formation and migration ↑ cleaved caspase-3, -9, Bax, p21, p27, T-cadherin, E-cadherin, ↓ Bcl-2, Cyclin D1, CdK4, MMP-2, MMP-9, PI3K, phospho-Akt, mRNA levels of OCT4, BMI1, SOX2, NANOG, NOTCH1, ABCG2, c-Myc, β-catenin, Dvl-2, LRP6, phospho-LRP6, Axin2, survivin, STAT3, phospho-STAT3, phospho-Jak2, phospho-MAPK, p300, CBP, Snail, Vimentin, phospho-Src, phospho-MEK, phospho-Smad2/3, phospho-S6	[114,115,116]
Rubone	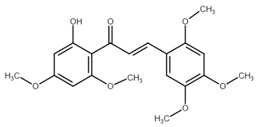	Inhibition of proliferation ↑ expression of miR23a ↓ Cyclin D1, Bcl-2	[117,118]
Xanthoangelol	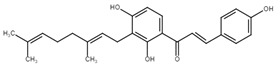	Induction of ER stress and G0/G1 phase arrest, inhibition of EMT and colony formation ↑ cleaved caspase-3, -8, -9, -12, PARP, Bak, Bax, cytosolic cytochrome c, Beclin 1, Atg5, LC3B-II, CHOP, GRP78, ATF6, p-eIF2α, IRE1α, phospho-JNK, phospho-c-jun, E-cadherin, phospho-AMPKα ↓ Bcl-2, mitochondrial cytochrome c, p62/SQSTM1, N-cadherin, vimentin, phospho-Akt, phospho-mTOR, phospho-RPS6KB1, activity of BRAF V600E	[88,119,120]
Flavokawains	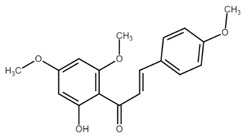	G2/M phase arrest, induction of autophagic cell death inhibition of tubulin polymerization and colony formation ↑ ROS, activity of GSTP1, cleaved caspase-3, -9, PARP, Bax, LC3 I/II, ATG7, p62/SQSTM1 p21, RhoA, H2AX, ↓ GSH, activity of GSS, Bcl-2, phospho-mTOR, Skp2, MMP-9	[121,122,123,124,125,126]
Poinsettifolin B	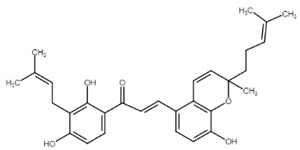	sub-G0/G1 phase arrest ↓ MMP ↑ ROS, cleaved caspase-3, -7, -8, -9	[127]
Lonchocarpin	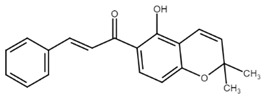	Inhibition of proliferation and migration ↓Bcl-2, nuclear β-catenin, activity of luciferase ↑ release of cytochrome c, Bax, cleaved caspase-3, -9	[128,129]
Millepachine	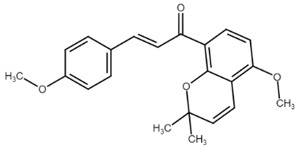	ATM, tubulin polymerization, G2/M phase arrest, ↓ expression of Bcl-2, Bcl-xl, TOPO II ↑ Bax, Bad, phospho-ATM, γ-H2AX,	[130,131,132]
Isocordoin and its analogues	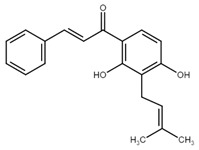	↓ Bcl-2, Bax, Hsp70 ↑ Bax, caspase-3, -9, ROS, DNA fragmentation	[133]
Derricin, derricidin, 4-hydroxyderricin	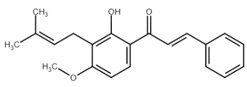	Inhibition of proliferation ↓ phospho-MEK, phospho-ERK, phospho-Akt, Cyclin D1, Bcl-2, activity of luciferase, β-catenin ↑ cleaved PARP, cleaved caspase-3	[119,134]
Phloretin	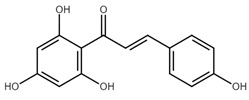	G0/G1 cycle arrest, inhibition of migration and colony formation ↓ Cyclin D1, Cyclin D2, CdK4, CdK6, phospho-Erk1/2, phospho-p38, phospho-JNK, MMP-2, MMP-3, Cathepsin S, CD44, Sox-2, VEGF ↑ LC3B II, Beclin 1, ROS,	[135,136,137]
4′,6′-dihydroxy-2′,4-dimethoxy-5′-(2″-hydroxybenzyl) dihydrochalcone	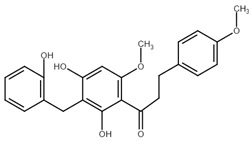	Inhibition of EGFR/MAPK pathway ↓ MMP, ↑ mitochondrial and cytosolic Ca^2+^, activity of caspase-3, -8, -9, Bim, Bid, Bad, ATR, ATM	[138]
Aspalathin	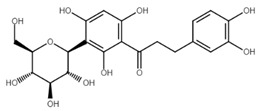	Suppression of migration and invasion, inhibition of YAP signaling, G2/M phase arrest ↓ Yap, MST1, Akt, phospho-Akt, paxillin, TRAF2, TRAF4, AVEN, PKM2, Mcl-1, CdK1, Bcl-xl, ↑ E-cadherin, cytochrome c, cleaved caspase-3, p21	[139,140]
Echinatin	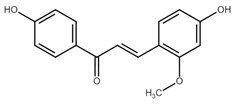	G2/M phase arrest, induction of ER stress ↓ Cyclin B1, Cdc2, Bcl-2, Mcl-1, Bcl-xl, mitochondrial cytochrome c, phospho-EGFR, phospho-MET, phospho-ERBB3, phospho-Akt, phospho-Erk ↑ p21, p27, ROS, CHOP, DR4, DR5, GRP78, phospho-JNK, Bax, phospho-p38, cytosolic cytochrome c, Apaf-1, cleaved PARP	[73,141]
Broussochalcone	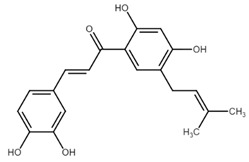	↓ Cyclin D1, c-Myc, Axin2, β-catenin, Bcl-2, phospho-Erk, phospho-Akt, ↑ cleaved PARP, cleaved caspase-3, FOXO3, p21, p27, p53, Bax,	[74,142]
Hydroxysafflor yellow A	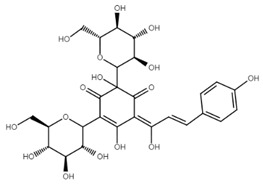	S phase cell cycle arrest ↓ Cyclin D1, Cyclin E, CdK2, PI3K, phospho-PI3K, Akt, phospho-Akt, phospho-Erk 1/2, Bcl-2, vimentin, N-cadherin, MMP-2, MMP-9, mTOR, ↑ LC3-II, Beclin 1, Bax, cleaved caspase-3, -9, E-cadherin	[75,143]
3-deoxysappanchalcone	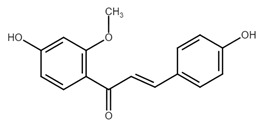	G2/M phase arrest ↓ Cyclin B1, TOPK, phospho-TOPK, phospho-Erk, phospho-RSK, c-Jun ↑ p53, p21, cleaved PARP, cleaved caspase-3, -7	[76]
Isobavachalcone	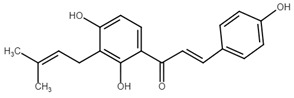	Increase of sub-G0/G1 phase, ER stress activation ↑ activity of caspase 3/7, -8, -9, levels of ROS, Bax, expression of GRP78, p-eIF2α, ATF4, XBP-1, Chop, phospho-β-catenin, ↓ MMP, Akt, phospho-Akt, Erk, phospho-Erk, Bcl-2, MMP-2, MMP-9, activity of TrxR1, XIAP, survivin, phospho-GSK-3β	[63,66,127,144]

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
