# Peer review of "Molecular Mechanisms of Antiproliferative Effects of Natural Chalcones"

_cancers, 2021, doi:10.3390/cancers13112730_

Round 1
Reviewer 1 Report
This is a robust review that is well-organized by category of the action of chalcones derivatives. This paper contains and summarizes many previously published studies. This paper shows the potential of natural chalcones as an antiproliferative agent. Therefore, I support its publication with minor comments.
- In the introduction, authors mentioned that anticancer agents induce severe organ toxicity and the natural compound has great interests. To support this, I would recommend that the information on the biosafety of natural chalcones compared to conventional anticancer drugs should be added.
- This paper introduces the diverse chalcone derivatives in Table 1. These molecules have various anticancer mechanisms depending on the type. Therefore, I recommend adding molecular structures to improve the utilization and understanding of these molecules.
- In the abbreviation section, some of the items don’t have space between abbreviation and original words.
- Line 215 “fytochemicals” should be “phytochemicals”
- endoplasmic reticulum in Line 241, 247, and 255 should be “ER” this abbreviation is in Line 238.
- Line 262 add original words of ROS.
- Line 525 add original words of MDM2 and then remove (mouse double minute 2) on line 526.
- Line 620 swap the positions of placental growth factor and PlGF.
Reviewer 2 Report
It is interesting topic. However the review should be partially rewrite. The review should be more condensed in information concern natural chalones and focus more on impact of natural chalcones on cellular process and molecular mechanisms/ pathways , avoiding long description of basic information, which easily can be find in many other reviews or books. The authors should shortage some text and instead cited review papers of other authors on, e.g. apoptosis or autophagy.
Figures are impressive, but I do not see the sense of showing molecular pathways without relations to chalcones. The figures should be removed or modified. I will expect to show in each figure the impact of chalcones, general or selected one, in the described processes, not just process itself.
I will avoid classification of type of cell death to 1,2,3, since there are many others known, and all of them in lead to cell death.
Reviewer 3 Report
no comments, very fine anad interesting
Round 2
Reviewer 2 Report
I have no further comments or suggestion.